# Analysis of Perioperative Platelet Indices and Their Prognostic Value in Head and Neck Cancer Patients Treated with Surgery and Postoperative Radiotherapy: A Retrospective Cohort Study

**DOI:** 10.3390/jcm8111858

**Published:** 2019-11-02

**Authors:** Bernhard J. Jank, Markus Haas, Daniela Dunkler, Nicholas J. Campion, Faris F. Brkic, Gregor Heiduschka, Lorenz Kadletz

**Affiliations:** 1Department of Otorhinolaryngology and Head and Neck Surgery, Medical University of Vienna, A-1090 Vienna, Austria; bernhard.jank@meduniwien.ac.at (B.J.J.); markus.haas01@icloud.com (M.H.); nicholas.campion@meduniwien.ac.at (N.J.C.); faris.brkic@meduniwien.ac.at (F.F.B.); gregor.heiduschka@meduniwien.ac.at (G.H.); 2Center for Medical Statistics, Informatics and Intelligent Systems, Medical University of Vienna, A-1090 Vienna, Austria; daniela.dunkler@meduniwien.ac.at

**Keywords:** head and neck squamous cell carcinoma, disease-free survival, overall survival, blood platelets, mean platelet volume, recurrence risk, biomarker

## Abstract

Objectives: Activated platelets might play an important role in tumor progression. Mean platelet volume (MPV) has been used as a surrogate marker for platelet activation, and therefore its value as a marker of tumor prognosis has attracted recent attention. In this study, we aimed to critically evaluate the prognostic significance of the perioperative platelet count (COP), MPV and the MPV/COP ratio in head and neck cancer patients. Additionally, we explored the individual postoperative trajectory of these indices and their association with overall survival (OS) and disease-free survival (DFS). Methods: We retrospectively evaluated 122 head and neck squamous cell carcinoma patients receiving surgery with curative intent followed by postoperative radiotherapy. Platelet indices were measured preoperatively and on days 1 and 7 postoperatively. OS and DFS were analyzed using Kaplan–Meier estimators, the log-rank test and uni and multivariable Cox models. Cutoffs to dichotomize patients for Kaplan–Meier curves and log-rank tests were empirically chosen at the respective median. The median follow-up was 8.8 years. Results: The adjusted preoperative COP, MPV and MPV/COP ratio were not associated with disease outcome. A low postoperative COP and a high MPV/COP ratio on the first postoperative day were independently associated with worse OS and DFS. In comparison to the preoperative measurements, patients whose COP increased by day 1 post-op showed a better OS (hazard ratio (HR) per 50 G/L increase: 0.73, 95% confidence interval (CI): 0.58–0.93, *p* = 0.013) and DFS (HR per 50 G/L increase: 0.74, 95% CI: 0.58–0.94, *p* = 0.018) in multivariable analysis. Conclusions: Our results suggest that a low postoperative COP and a high MPV/COP ratio represent a negative prognostic factor for OS and DFS. Notably, patients with an increase in COP by day 1 post-op when compared to their preoperative value showed a significantly better OS and DFS.

## 1. Introduction

Head and neck squamous cell carcinoma (HNSCC) account for 63,500 annual deaths, with approximately 250,000 new cases every year in Europe alone [1]. HNSCCs are mainly associated with alcohol and tobacco use; however, more recently, the proportion of HNSCC cases attributable to human papillomavirus (HPV) infections has risen sharply. These tumors affect younger patients and have a more favorable prognosis compared to HNSCCs with other etiologies [1,2]. Curative treatment approaches for early-stage HNSCCs mainly consist of single-modality surgery or radiotherapy alone. Late-stage HNSCCs are managed by surgery, followed by postoperative radiotherapy (PORT) or escalated to chemoradiotherapy for patients with extracapsular extension or R1 resection at surgery. [3,4] The eighth edition of TNM staging for oropharyngeal squamous cell carcinoma (OPSCC) includes immunohistochemical staining for p16 to identify OPSCCs associated with high-risk HPV. This highlights the importance of biomarkers for the further risk stratification of mortality risk in HNSCC staging and prognosis [5].

In their quiescent form, platelets are small, anucleated cells which are derived from megakaryocytes and which circulate in the blood [6]. Besides their physiologic role in hemostasis and innate immunity, it has been experimentally shown that activated platelets also play an important role in all steps of tumorigenesis [7]. For example, platelets can build a physical barrier, shielding tumor cells from natural killer cells [8] as well as tumor necrosis factor α [9]. Furthermore, vascular endothelial growth factor (VEGF) is an important angiogenic protein, and platelets are the major circulating source of VEGF [10]. Also, the secretion of transforming growth factor ß (TGF-ß) in the tumor microenvironment by platelets has been shown to induce genes that initiate epithelial-to-mesenchymal transition genes and thereby increase tumor invasiveness [11,12]. Contrarily, platelets can also exert antitumor effects by transferring miRNAs via platelet-derived microparticles into solid tumors, which results in the downregulation of oncogenes [13]. This recent study demonstrated that platelets can therefore also contribute to tumor suppression, highlighting the complexity of the platelet–cancer interaction.

Mean platelet volume (MPV) has been used as a surrogate marker for platelet activation in several studies [14,15]. While further markers such as P-selectin [16] or active GPIIb/IIIa [17] have been identified, they require more complex analysis and are not yet feasible for routine clinical use.

MPV has been shown to be a prognostic factor in various malignant diseases, such as colorectal cancer [18,19,20,21], pancreatic cancer [22], lung cancer [23], and others [15]. Notably, further studies have shown an inverse relationship between platelet count (COP) and MPV, which subsequently led to recent studies analyzing platelet indices as a ratio [24,25,26]. Notably, no study to date has analyzed the postoperative trajectory of platelet indices and their association with overall survival and disease-free survival in HNSCC. Cancer surgery causes an acute inflammatory response, made up of two parts: an acute pro-inflammatory phase, followed by an anti-inflammatory phase [27]. These two processes involve a complex cascade of events which includes the activation of platelets with the release of many growth factors, chemokines and cytokines, which may also promote the tumor progression of residual or distant cancer cells [28]. The aim of this retrospective study was therefore to determine the trajectory of platelet-associated markers pre and postoperatively, as well as how they are associated with overall and disease-free survival in HNSCC patients.

## 2. Patients and Methods

### 2.1. Study Design & Patient Population

This study is a single-center, observational, retrospective cohort study. The study population was drawn from our in-house retrospective head and neck squamous cell cancer cohort, which includes 130 patients treated with curative surgical therapy followed by postoperative radiotherapy. Exclusion criteria were external treatment, secondary primary carcinoma and prior irradiation. All patients were histologically diagnosed HNSCC patients who were treated at the Medical University of Vienna with surgery and postoperative irradiation between 2002 and 2012. The last follow-up occurred in September 2018. Baseline and outcome data were collected retrospectively from electronic patient records. The records included the following clinical data: age at diagnosis, sex, tumor site, HPV high-risk status, TNM classification in accordance with eighth edition American Joint Committee on Cancer (AJCC) staging, disease-free survival, smoking status, alcohol consumption, radiation dose received, administered chemotherapy, platelet count and mean platelet volume. HPV assessment was performed with in-situ hybridization as described previously [29]. In brief, HPV detection was carried out with a validated detection system (Ventana; INFORM^®^ Probes In Situ Hybridization (ISH) system) containing a mixture of DNA probes specific for HPV high-risk genotypes 16, 18, 31, 33, 35, 39, 45, 51, 52, 56, 58, and 66. HPV in situ hybridization staining was analyzed according to the interpretation guidelines (positive = 1, negative = 0) provided by the manufacturer. For preoperative values, we included laboratory data that had been collected within a timeframe of at most 2 weeks before surgery. Post-op lab data was collected on day 1 and 1 week (7 ± 2 days) after the day of surgery. Platelet count (COP) preoperative laboratory data could be extracted for 122 patients and were measured in Giga/L (=10^9^/L). Preoperative MPV (in femtoliter, FL) was available for 113 patients. Postoperatively, COP data could be extracted for 112 and 110 patients on postoperative days 1 and 7, respectively. MPV data was available for 112 and 109 patients on post-op days 1 and 7, respectively. The MPV/COP ratio was calculated by dividing MPV (in FL) by COP (in G/L) * 100. To analyze each patient’s individual postoperative platelet trajectory, we calculated the difference for each post-op timepoint by subtracting the pre-op value from the respective post-op value. Values > 0 were scored as an increase and values < 0 as a decrease. This study was approved by the institutional research board (ECS 1311/2018).

### 2.2. Statistical Analysis

The primary outcome of this study was overall survival (OS). The secondary outcome was disease-free survival (DFS). In the absence of validated cut-offs for all markers, we pre-specified to select empirical cutoffs at the median values of the platelet indices at each respective timepoint, and patients were dichotomized into either high (>median (Q2)) or low (≤median (Q2)) marker groups accordingly (necessary for Kaplan–Meier curves and log-rank test). Continuous variables were summarized as medians (25th to 75th percentile = interquartile range, IQR) and categorial variables were summarized as absolute counts and percentages (%). Categorical baseline variables were compared using Chi^2^ or Fisher’s exact test, and continuous baseline variables were compared using t-test or Wilcoxon rank-sum test, depending on the distribution. Follow-up was defined as the time from the day of surgery until recurrence, death or censoring alive. Median follow-up time was estimated with a reverse Kaplan–Meier estimator according to Schemper et al. [30]. Rates for OS were analyzed using the Kaplan–Meier estimator, and DFS was analyzed using the Kaplan–Meier failure function. Differences between groups were compared using the log-rank test.

Univariable and multivariable analyses were performed using the Cox proportional hazard regression. Hazard ratio (HR) and 95% confidence intervals (95% CI) were calculated per 50 G/L increase for COP, per 1 FL increase in platelet volume for MPV and per 1 fraction for the MPV/COP ratio. For postoperative platelet indices on days 1 and 7, time zero in the Kaplan–Meier and Cox analysis was either day 1 or day 7. A two-sided *p*-value < 0.05 was considered as statistically significant. Statistical analyses were performed using Stata (Macintosh version 16.0, Stata Corp, Houston, TX, USA).

## 3. Results

### 3.1. Analysis at Baseline

One-hundred and twenty-two patients diagnosed with HNSCC were included in this retrospective cohort study. Baseline data of the patient population can be seen in Table 1.

HPV status was known for 116 patients, of which 25 patients (22%) tested positive for high-risk HPV genotypes using in situ hybridization. All patients underwent surgery with curative intent, followed by postoperative radiotherapy, with a dose ranging from 40–70 Gy. Eleven (12%) patients received additional postoperative chemotherapy because of extracapsular spread or R1 resection in the final histopathologic workup. Smoking status was known for 121 patients, of which 67% (*n* = 81) were active smokers. Alcohol consumption was known for 107 patients, of which 39% (*n* = 42) were active drinkers. Sixteen patients (13%) received antiplatelet agents (acetylsalicylic acid), which were paused at least 10 days prior to surgery. During a median follow-up of 8.8 years (range: 6.3–11.7 years), we observed 59 (48%) deaths and 42 (34%) disease recurrences. The 5-year OS and DFS rates were 54% (95% CI: 45–62) and 49% (95% CI: 40–58), respectively.

### 3.2. Preoperative Platelet Indices and Time to Event Analysis

The median preoperative COP and MPV values were 237 G/L (191–302) and 10.3 FL (9.8–11.1), respectively, and the median MPV/COP ratio was 4.3 (3.3–5.4) (Figure 1).

High preoperative COP, MPV and MPV/COP values did not predict worse OS (Figure 2, top row) or DFS (Appendix A, top row). In univariable Cox regression, COP, MPV and the MPV/COP ratio were not associated with OS or DFS (Appendix A). These results remained unchanged in multivariable analysis after adjusting for TNM stage, HPV status and smoker status.

### 3.3. Postoperative Platelet Indices and Time to Event Analysis

We assessed the platelet indices after surgical intervention. We observed changes in the pre and postoperative COP and MPV/COP ratio (Figure 1). The median COP and MPV on day 1 post-op were 176 G/L (142–225.5) and 10.5 (9.65–11.3) and at one week post-op were 286 G/L (218–346) and 10.2 fl (9.65–11), respectively. The calculated MPV/COP ratio was 6.07 (4.48–7.80) on day 1 and 3.6 (2.8–4.8) at one week post-op.

The mean COP decreased on day 1 post-op compared to the preoperative value (median (IQR): 237 (191–302) vs. 174 (140–225), *p* < 0.001) with a median decrease of −61 G/L (−88.5 to −27). In contrast, the COP value at 1 week post-op increased in comparison the preoperative value (median (IQR): 237 (191–302) vs. 286 (218–346), *p* < 0.001) with a median increase of 37.5 G/L (−3 to 88) (Figure 1). Compared to preoperative MPV, we observed no difference in MPV on day 1 post op (median (IQR) in FL: 10.3 (9.8–11.1) vs. 10.5 (9.65–11.3), *p* = 0.344) and no difference at one week post op (median (IQR) in FL: 10.3 (9.8–11.1) vs. 10.2 (9.65–11), *p* = 0.311) (Figure 1). The MPV/COP ratio increased accordingly on day 1 post-op (median (IQR): 4.3 (3.3–5.5) vs. 6.1 (4.5–7.8), *p* < 0.001) and decreased at one week post-op (4.3 (3.3–5.5) vs. 3.6 (2.9–4.8), *p* < 0.001) with a median change of 1.5 (0.8–2.2) and −0.70 (−1.53 to 0.05), respectively (Figure 1).

A low postoperative COP showed a numerically worse OS at all timepoints, which was associated with OS at one week post-op (5-year OS for COP ≤ Q2: 49% (95% CI 35–62) vs. COP > Q2: 74% (95% CI 60–83)) (Figure 2).

Furthermore, we observed a numerically worse OS for patients with a high MPV/COP ratio, with a significant difference at one week post-op (5-year OS for MPV/COP > Q2 ratio 48% (95% CI 34–61) vs. MPV/COP ≤ Q2 75% (95% CI 61–84)).

In univariable Cox regression, we observed an association for COP at day 1 post-op and OS (HR per 50 G/L increase: 0.78, 95% CI: 0.62–0.99, *p* = 0.046). This result prevailed after adjustment for smoker status, TNM stage and HPV status (HR: 0.76, 95% CI: 0.60–0.95, *p* = 0.020). Furthermore, we observed an association of the MPV/COP ratio and OS (HR per one fraction increase: 1.13, 95% CI: 1.03–1.24, *p* = 0.006) on day 1 post-op. This result also prevailed after adjustment for smoker status, TNM stage and HPV status in multivariable analysis (HR: 1.12, 95% CI: 1.03–1.23, *p* = 0.009) (Table 2).

Patients with low postoperative COP at one week showed a worse DFS (5-year DFS for COP ≤ Q2: 62% (95% CI 46–74) vs. COP > Q2: 76% (95% CI 62–86), log-rank *p* = 0.019). A high postoperative MPV/COP ratio at one week also showed a worse DFS (5-year DFS for MPV/COP ≤ Q2: 76% (95% CI 62–86) vs. MPV/COP > Q2: 62% (95% CI 47–74), log-rank *p* = 0.026) (Appendix A). In univariable analysis for DFS, we observed differences for postoperative COP on day 1 (HR per 50 G/L increase: 0.70, 95% CI 0.52–0.94, *p* = 0.021) and one week (HR per 50 G/L increase: 0.80, 95% CI 0.66–0.99, *p* = 0.040) and MPV/COP ratio on day 1 (HR per 1 fraction increase: 1.13, 95% CI 1.01–1.27, *p* = 0.022) and one week (HR per 1 fraction increase: 1.26, 95% CI 1.05–1.51, *p* = 0.013). This result prevailed in multivariable analysis after correction for TNM stage, HPV status and smoker status (Table 2). We could not observe any association in OS or DFS with postoperative MPV.

### 3.4. Postoperative Trajectory of Platelet Indices

We analyzed the association of the individual postoperative marker trajectory on OS and DFS. On day 1 post-op, the COP decreased in 101 patients (90%), while it increased in 11 (10%) patients compared to the pre-op baseline. At one week post-op, it decreased in 31 patients (28%), while it increased in 79 patients (72%). The MPV decreased in 65 patients (63%) and 69 (67%) and increased in 38 (37%) and 34 (33%) patients on days 1 and 7, respectively. The MPV/COP ratio decreased in 9 (9%) and 75 (73%) patients and increased in 94 (91%) and 28 (27%) patients on days 1 and 7, respectively (Figure 1) (Appendix A). While the MPV trajectory was not associated with OS or DFS, an increase in COP on day 1 post-op was associated with better OS in univariable analysis (HR per 50 G/L increase: 0.73, 95% CI: 0.59–0.90, *p* = 0.004). This result prevailed in multivariable analysis (HR per 50 G/L increase: 0.73, 95% CI: 0.58–0.93, *p* = 0.013) (Figure 3A, Table 3).

An increase in COP on day 1 was also associated with a better DFS in univariable (HR per 50 G/L increase: 0.72, 95% CI: 0.56–0.91, *p* = 0.006) and multivariable analysis (HR per 50 G/L increase: 0.74, 95% CI: 0.58–0.94, *p* = 0.018) (Figure 3B, Table 3). All HR values are summarized in Figure 4.

The group of patients with increasing COP (11 of 112) on the first postoperative day included significantly more female patients (21% female (*n* = 6) vs. 6% male (*n* = 5), *p* = 0.017). There was no difference in MPV between patients with increasing or decreasing COP on day 1 post-op (10.1 (9.4–11.8) vs. 10.5 (9.8–11.2), *p* = 0.903). No other associations with baseline characteristics could be found. For the MPV/COP ratio, an increase from baseline by day 1 post-op was associated with a worse OS in univariable (HR: 1.20, 95% CI: 1.06–1.35, *p* = 0.003) and multivariable analysis (HR: 1.18, 95% CI: 1.04–1.34, *p* = 0.007) as well as for DFS in univariable (HR: 1.21, 95% CI: 1.04–1.41, *p* = 0.011) and multivariable analysis (HR: 1.17, 95% CI: 1.01–1.37, *p* = 0.035).

## 4. Discussion

In addition to their physiological role in hemostasis, coagulation and innate immunity, growing evidence suggests an interaction between platelets and tumor cell growth and metastasis [31]. Previous studies have analyzed the potential role of preoperative platelet count and MPV on disease outcomes in different cancer entities.

MPV has been described as a prognostic factor in numerous tumor entities, such as lung cancer [7], gastric cancer [32], colorectal cancer [18] and others [15]. Notably, no study to date has directly investigated an association between MPV and disease outcome in HNSCC patients treated with surgery and PORT. However, an elevated preoperative platelet count has been associated with poor prognosis in a recent meta-analysis in HNSCC patients [33], and a recent publication by Park et al. investigated the COP–MPV score as prognostic factor for disease outcome in patients with oral squamous cell carcinoma (OSCC) [34]. The significant association between the COP–MPV score and disease outcome described for OSCC, however, could not be validated by Tham et al. in a validation study in HNSCC patients [35]. Notably, Park et al. determined the cutoffs to calculate the COP–MPV score of a relatively small cohort by ROC analysis without the use of a validation cohort and might therefore have overfitted the model.

In this study, we aimed to elucidate the trajectory of perioperative platelet indices and analyze their effect on disease outcomes. For preoperative COP, MPV and the MPV/COP ratio, we could not find any association with OS or DFS. In contrast, the above-mentioned meta-analysis, published in 2018 by Takenaka et al., selected six studies and came to an overall conclusion that a higher pre-treatment platelet count was associated with a worse OS in HNSCC [33]. However, the cutoff values of those studies were inconsistent and derived by different methods with a wide variety (150–400 G/L); thus, the pooled HRs are difficult to interpret [33]. Most importantly, the number of reported negative results was fewer than expected, and therefore publication bias could be a significant factor [33].

Tissue trauma caused by cancer surgery is a major cause of platelet activation. We hypothesized that a patient’s individual platelet response and therefore the postoperative trajectory might differ, and that this difference might influence disease outcome. Data on post-treatment platelet indices are sparse. A recent publication by Qian et al. investigated post-treatment platelet-associated factors in patients with resectable colorectal cancer [19]. Post-treatment blood samples were collected 3 weeks after surgery and at 3 months after adjuvant chemotherapy. Results showed that pre-treatment COP and MPV did not correlate with outcomes. The authors found, however, that a decrease in post-treatment MPV was associated with a poorer OS, while they did not find any associations for COP, plateletcrit or platelet distribution width [19]. A study by Zhong et al. [36] analyzed the postoperative change in the platelet to lymphocyte ratio (PLR)—a marker for systemic inflammation—in T3–T4 laryngeal squamous cell carcinoma. They found that a postoperative increase in PLR was independently associated with worse OS and DFS.

In this study, we analyzed the postoperative platelet indices on day 1 and one week post-op and their association with disease outcomes. Surgery significantly decreased the COP on the first postoperative day, while it increased after one week in the majority of patients. Interestingly, we could not find any significant differences between pre and postoperative MPV. Compared to the study by Qian et al., we measured MPV early in the postoperative phase and might therefore have missed a lagged phase of change in MPV. Furthermore, we could not find any association between postoperative MPV and disease outcome. For postoperative COP, however, we observed a worse OS and DFS for patients with low COP (≤ median), with a risk reduction of 24% for OS and 31% for DFS per 50 G/L increase in COP at day 1 post-op. We then investigated the individual trajectory of platelet indices and found that none of the 11 patients whose COP increased on the first postoperative day died during follow up, although this was not significant in log-rank test, most likely because of the small number of observations (*n* = 11). We found an independent association between increasing COP on the first postoperative day and OS as well as DFS in multivariable Cox regression. Besides the presence of significantly more female patients in this group, we could not find any association with other baseline characteristics. We hypothesize that platelet activation through surgical trauma is either greatly reduced in these patients, or the bone marrow response might be substantially enhanced. Interestingly, Michael et al. recently demonstrated that platelets are also capable of suppressing tumor growth by transferring platelet-derived RNA by infiltrating platelet-derived microparticles into solid tumors [13]. This mechanism could potentially explain the better survival of patients with an increasing COP post-op, if they are capable of exerting an anti-tumor response. Contrarily, a low COP post-op could be caused by a mechanism called “tumor-educated platelets”. This concept suggests a crosstalk between cancer cells and platelets with increased platelet activation and aggregation, resulting in protumor behavior and subsequently fewer circulating platelets [37]. However, the mechanisms underlying this finding remain speculative, especially considering the complex process of platelet maturation and activation. For the individual MPV trajectory, notably, we could not find any association with disease outcome. Therefore, our data strongly suggests no association between pre and postoperative MPV or the trajectory of postoperative MPV and disease outcome. These results are in line with the conclusion of a recent meta-analysis by Pyo et al., who could not find any association between MPV and survival in different malignant diseases [15]. Notably, MPV is not a reliable marker for platelet activation and should therefore be interpreted with caution [38].

Finally, we analyzed the association of the MPV/COP ratio on disease outcome. The MPV/COP ratio also showed an independent association with OS on the first postoperative day and with DFS at day 1 and 1 week postoperatively, which was comparable to that observed with COP. We could not find a significance of the MPV/COP ratio over COP in predicting disease outcomes in our study.

There are several limitations of this study that need to be discussed. First, this study was a single-center study with a small study population and a retrospective study design. Selection bias can therefore not be fully excluded. However, we used a homogenously treated patient population to limit this potential bias. Second, we did not exclude patients with other systemic diseases—i.e., diabetes, hypertension or vascular disease—that are known to have an impact on platelet indices. However, our intention was to analyze platelet indices as potential prognostic biomarkers for routine clinical use, and therefore our study population reflects the clinical profile of the HNSCC patient population. Notably, no platelet function assessment was performed. However, since all patients were fit for major cancer surgery, severely altered platelet function is unlikely. Furthermore, the median preoperative values used as cutoffs in this study were within the normal laboratory range for COP and MPV and comparable to the cutoffs used in previous studies for COP [33] and MPV [15]. Finally, we did not evaluate the duration of surgery and the intraoperative blood loss. Although a correlation between later TNM stages with longer surgery and increased blood loss and a subsequent decrease in COP and increase in MPV seems likely, we did not find any significant differences between postoperative COP or MPV and TNM-stage (data not shown).

In conclusion, the prognostic role of preoperative COP and the MPV/COP ratio in HNSCC patients treated with surgery and PORT remains unclear. Based on our study results, MPV is not a useful biomarker in HNSCC patients. However, our results do suggest that low COP during the first postoperative week might be a risk factor for worse overall survival and disease-free survival. In particular, patients with an increase in COP on the first postoperative day showed superior survival.

## Figures and Tables

**Figure 1 jcm-08-01858-f001:**
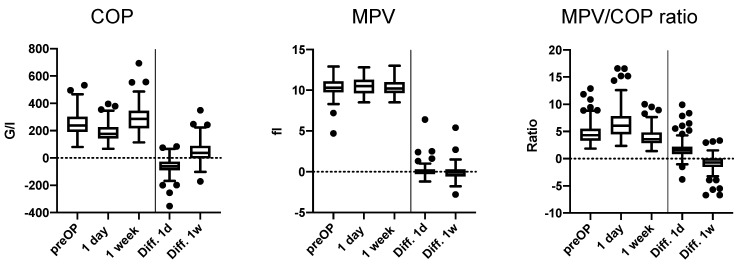
Tukey boxplots for pre and postoperative platelet indices and the differences in perioperative platelet count (COP), mean platelet volume (MPV) and the MPV/COP ratio on 1 day and 1 week post-op compared to the pre-operative measurements.

**Figure 2 jcm-08-01858-f002:**
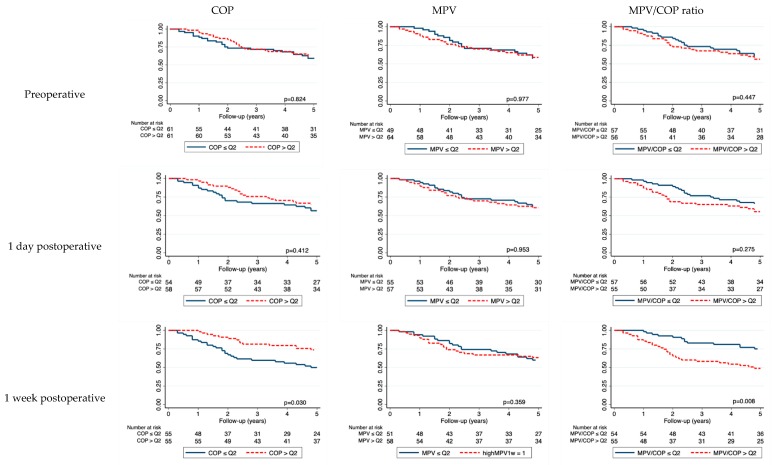
Kaplan–Maier curve according to preoperative (top rows) and postoperative (bottom rows) platelet indices for overall survival (OS). Y axis, surviving fraction. Abbreviations: Q2, median.

**Figure 3 jcm-08-01858-f003:**
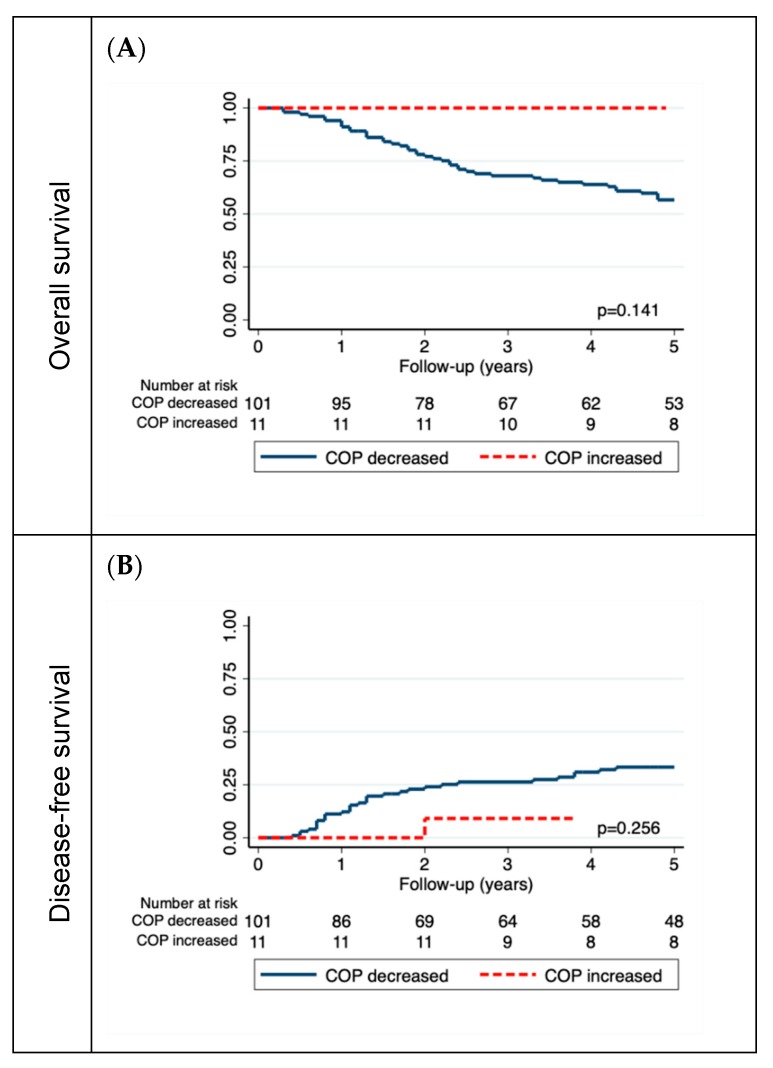
(**A**) Kaplan–Meier curve for overall survival (OS) comparing patients where COP decreased or increased from the preoperative measurement to the postoperative measurement on day 1, and (**B**) Kaplan–Meier failure function for disease-free survival (DFS).

**Figure 4 jcm-08-01858-f004:**
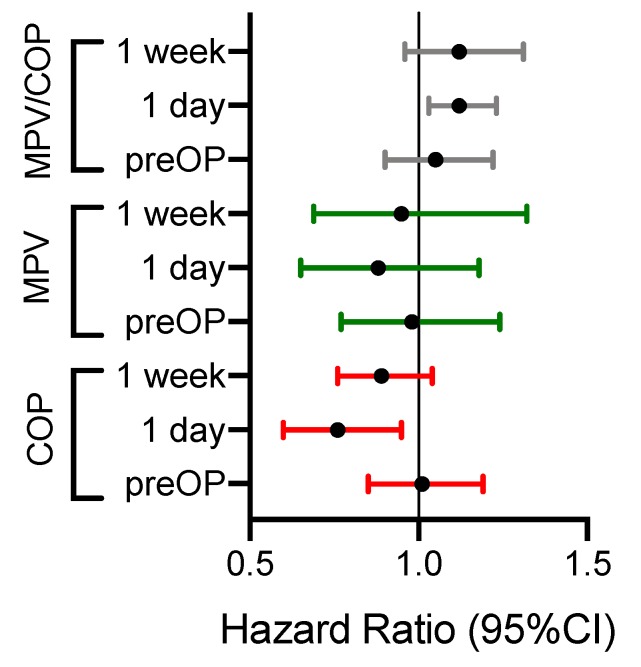
Forest plots of hazard ratios (HR) and 95% CI for overall survival of the platelet indices at preoperative and postoperative measurements (HR: COP: per 50 G/L increase; MPV: per 1 FL increase, MPV/COP ratio: per 1 fraction).

**Table 1 jcm-08-01858-t001:** Baseline characteristics of patients included in the study.

	Total (*n* = 122)
Gender	
Female	29 (24%)
Male	93 (76%)
Age at diagnosis	59
(Q1–Q3)	(52–63)
T-classification	
T1	26 (21%)
T2	65 (53%)
T3	20 (16%)
T4	11 (9%)
N-classification	
N0	25 (20%)
N1	21 (16%)
N2a	6 (5%)
N2b	31 (25%)
N2c	10 (8%)
N3	1 (1%)
N3b	3 (2%)
pN0 (HPV+)	1 (1%)
pN1 (HPV+)	6 (5%)
pN2 (HPV+)	16 (13%)
pN3 (HPV+)	2 (2%)
TNM-Staging	
I	10 (8%)
II	29 (24%)
III	23 (19%)
IVA	56 (46%)
IVB	4 (3%)
HPV	*n* = 116
-	91 (78%)
+	25 (22%)
Primum	
Hypopharynx	20 (16%)
Larynx	14 (12%)
Oral Cavity	32 (26%)
Oropharynx	58 (46%)
Alcohol consumption	*n* = 107
Non-drinker	65 (61%)
Active drinker	42 (39%)
Smoking status	*n* = 121
Non/Ex-Smoker	40 (33%)
Smoker	81 (67%)
Platelet inhibitor	
No	106 (87%)
Yes	16 (13%)

HPV: human papillomavirus.

**Table 2 jcm-08-01858-t002:** Postoperative platelet indices: Univariable and multivariable time-to-event analysis.

		Univariable			Multivariable		
		HR	95% CI	*p*-Value	HR	95% CI	*p*-Value
**Overall Survival**							
**1 day**	COP (per 50 G/L increase)	0.78	0.62–0.99	0.046	0.76	0.60–0.95	0.020
	MPV (per 1 FL increase)	0.94	0.70–1.25	0.696	0.88	0.65–1.18	0.415
	MPV/COP ratio per 1 increase	1.13	1.03–1.24	0.006	1.12	1.03–1.23	0.009
**1 week**	COP (per 50 G/L increase)	0.90	0.77–1.04	0.177	0.89	0.76–1.04	0.172
	MPV (per 1 FL increase)	1.04	0.76–1.42	0.768	0.95	0.69–1.32	0.800
	MPV/COP ratio per 1 increase	1.14	0.97–1.34	0.092	1.12	0.96–1.31	0.145
**Disease-Free Survival**							
**1 day**	COP (per 50 G/L increase)	0.70	0.52–0.94	0.021	0.69	0.52–0.92	0.014
	MPV (per 1 FL increase)	0.94	0.66–1.33	0.747	0.98	0.68–1.41	0.951
	MPV/COP ratio per 1 increase	1.13	1.01–1.27	0.022	1.12	1.01–1.25	0.026
**1 week**	COP (per 50 G/L increase)	0.80	0.66–0.99	0.040	0.81	0.66–0.99	0.045
	MPV (per 1 FL increase)	1.03	0.70–1.51	0.862	1.02	0.69–1.52	0.884
	MPV/COP ratio per 1 increase	1.26	1.05–1.51	0.013	1.22	1.02–1.46	0.026

The four multivariable models were adjusted for TNM stage, HPV status and smoker status. HR: hazard ratio.

**Table 3 jcm-08-01858-t003:** Change in postoperative COP compared to the preoperative value: univariable and multivariable time-to-event analysis.

	Univariable			Multivariable		
	HR	95% CI	*p*-Value	HR	95% CI	*p*-Value
**Overall Survival**						
1 day	0.73	0.59–0.90	0.004	0.73	0.58–0.93	0.013
1 week	0.87	0.72–1.06	0.178	0.87	0.70–1.07	0.196
**Disease-Free Survival**						
1 day	0.72	0.56–0.91	0.006	0.74	0.58–0.94	0.018
1 week	0.80	0.62–1.02	0.083	0.80	0.62–1.04	0.102

The four multivariable models were adjusted for TNM stage, HPV status and smoker status. Hazard ratios (HR) and 95% confidence intervals were given per 50 G/L increase of the difference in COP at day 1 or week 1 compared to the preoperative measurement.

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
