# Peer review of "Analysis of Perioperative Platelet Indices and Their Prognostic Value in Head and Neck Cancer Patients Treated with Surgery and Postoperative Radiotherapy: A Retrospective Cohort Study"

_jcm, 2019, doi:10.3390/jcm8111858_

Round 1
Reviewer 1 Report
In the manuscript entitled “Analysis of perioperative platelet indices and their prognostic value in head and neck cancer patients treated with surgery and postoperative radiotherapy: a retrospective cohort study”, Bernhard J. Jank and coworkers evaluated the prognostic significance of the perioperative platelet count (COP), mean platelet volume (MPV) and the MPV/COP ratio in head and neck cancer patients. They also studied the individual postoperative trajectory of these indices and their association with overall survival (OS) and disease-free survival (DFS). The authors showed that a low postoperative COP and a high MPV/COP ratio, represent a negative prognostic factor for OS and DFS. This reviewer feel that the manuscript is interesting. However, there are some concerns that authors should address (mainly manuscript writing but not more experiments) to further improve its quality/safety before its publication:
The authors suggested that a low postoperative COP and a high MPV/COP ratio, represent a negative prognostic factor for overall survival (OS) and disease-free survival (DFS). These are interesting observations. However, as the authors are aware, they did not evaluate the duration of surgery and the intraoperative blood loss. The severe/progressive/bigger/late stage cancers will decrease overall survival (OS) and disease-free survival (DFS), and these patients will need longer time for surgery and have likely more blood loss, which will decrease platelet counts (COP) that may further drive new platelet generation (i.e. MPV increases). The authors should clearly introduce this limitation to the readers.
There is also no clear explanation/discussion why a low postoperative COP and a high MPV/COP ratio, represent a negative prognostic factor for OS and DFS. Is this due to the decreased platelets anti-cancer activity (reference: Michael JV,et al. Blood. 2017;130(5):567-580) or due to platelet-cancer cell adhesion that decreased the platelet count in the peripheral blood and enhanced tumor metastasis/growth (reference: Xu XR et al., Blood. 2018 Apr 19;131(16):1777-1789)? The authors should read/cite these published papers and provide more information for the readers.
In the abstract, the authors introduced “Mean platelet volume (MPV) is a marker for platelet activation”, which should be used with caution. Although the MPV may have some values as a predictor of outcomes in different cancer entities, it is not a good platelet activation marker. High MPV may reflect more new platelets in the blood circulation, and young platelets (bigger) may have higher functionalities. Platelet surface P-selectin (CD62P; reference: Yang H et al., Blood. 2009 Jul 9;114(2):425-36.), active GPIIbIIIa (reference: Reheman A et al., Circ Res. 2014 Mar 28;114(7):1070-3.) as well as other alpha and dense granule release are the markers for platelet activation (reference: Yang Y et al., PLoS One. 2012;7(5):e37323). The authors may read/cite some of these publications in their Introduction and Discussion, and correct their abstract before its publication.
In the introduction (page 2, line 53). The authors may read/cite the review article regarding the platelet versatilities (reference: Xu XR et al. Crit Rev Clin Lab Sci. 2016 Dec;53(6):409-30.); In the results section (page 4, line 130), the authors may list some of these anti-platelet agents for readers.
The Results section (page 4, line 137-138), the figure legend 1 is too simple to understand the figure. What are the “Diff”? Why is there no dot in the MPV for the 1 day and 1 week? There is also no sufficient information for Y axis. The sufficient information in the figure legend will markedly help the readers to understand their findings.
The same is also true for the over simplified figure legend 2, which is difficult for readers to understand the information from these 9 figures.
The authors may consider to decrease usage some of the abbreviations since they increase the difficulties for readers to understand their findings in the manuscript.
Author Response
Reviewer 1:
In the manuscript entitled “Analysis of perioperative platelet indices and their prognostic value in head and neck cancer patients treated with surgery and postoperative radiotherapy: a retrospective cohort study”, Bernhard J. Jank and coworkers evaluated the prognostic significance of the perioperative platelet count (COP), mean platelet volume (MPV) and the MPV/COP ratio in head and neck cancer patients. They also studied the individual postoperative trajectory of these indices and their association with overall survival (OS) and disease-free survival (DFS). The authors showed that a low postoperative COP and a high MPV/COP ratio, represent a negative prognostic factor for OS and DFS. This reviewer feel that the manuscript is interesting. However, there are some concerns that authors should address (mainly manuscript writing but not more experiments) to further improve its quality/safety before its publication:
Response: We thank reviewer #1 for the time spent to review our manuscript and the valuable and insightful comments that help us to improve our study. It is clear from the depth of the comments that the reviewer is an expert in the field.
The authors suggested that a low postoperative COP and a high MPV/COP ratio, represent a negative prognostic factor for overall survival (OS) and disease-free survival (DFS). These are interesting observations. However, as the authors are aware, they did not evaluate the duration of surgery and the intraoperative blood loss. The severe/progressive/bigger/late stage cancers will decrease overall survival (OS) and disease-free survival (DFS), and these patients will need longer time for surgery and have likely more blood loss, which will decrease platelet counts (COP) that may further drive new platelet generation (i.e. MPV increases). The authors should clearly introduce this limitation to the readers.
Response: Again, we thank the reviewer for the comment. Indeed, as we stated under limitations in the discussion, we did not evaluate duration of surgery and intraoperative blood loss. However, we analyzed the TNM-stage as a surrogate parameter for the probable extend of surgery and found no association with postoperative platelet count at one day (Chi2 p=0.212) or one week (Chi2 p=0.364) or MPV at one day or one week, respectively (Chi2 p=0.655 and p=0.800) (data not shown in the manuscript). We added this to the discussion. Also, TNM-stage is included in the multivariable Cox analysis as a possible confounder.
There is also no clear explanation/discussion why a low postoperative COP and a high MPV/COP ratio, represent a negative prognostic factor for OS and DFS. Is this due to the decreased platelets anti-cancer activity (reference: Michael JV,et al. Blood. 2017;130(5):567-580) or due to platelet-cancer cell adhesion that decreased the platelet count in the peripheral blood and enhanced tumor metastasis/growth (reference: Xu XR et al., Blood. 2018 Apr 19;131(16):1777-1789)? The authors should read/cite these published papers and provide more information for the readers.
Response: We thank the reviewer for the suggested literature. Indeed, the mechanisms in the above-mentioned publications could theoretically explain our observation. We included this in the discussion.
In the abstract, the authors introduced “Mean platelet volume (MPV) is a marker for platelet activation”, which should be used with caution. Although the MPV may have some values as a predictor of outcomes in different cancer entities, it is not a good platelet activation marker. High MPV may reflect more new platelets in the blood circulation, and young platelets (bigger) may have higher functionalities. Platelet surface P-selectin (CD62P; reference: Yang H et al., Blood. 2009 Jul 9;114(2):425-36.), active GPIIbIIIa (reference: Reheman A et al., Circ Res. 2014 Mar 28;114(7):1070-3.) as well as other alpha and dense granule release are the markers for platelet activation (reference: Yang Y et al., PLoS One. 2012;7(5):e37323). The authors may read/cite some of these publications in their Introduction and Discussion, and correct their abstract before its publication.
Response: We made the according changes in the abstract, introduction and discussion.
In the introduction (page 2, line 53). The authors may read/cite the review article regarding the platelet versatilities (reference: Xu XR et al. Crit Rev Clin Lab Sci. 2016 Dec;53(6):409-30.); In the results section (page 4, line 130), the authors may list some of these anti-platelet agents for readers.
Response: We thank the reviewer for suggesting the above-mentioned review. We added a reference in the introduction. We added the antiplatelet agent as recommended.
The Results section (page 4, line 137-138), the figure legend 1 is too simple to understand the figure. What are the “Diff”? Why is there no dot in the MPV for the 1 day and 1 week? There is also no sufficient information for Y axis. The sufficient information in the figure legend will markedly help the readers to understand their findings.
Response: We added an explanation for the calculation of the difference to the methods section.
The dots in the Tukey method plots represent values that are greater than the 75th percentile plus 1.5 times interquartile range. For MPV at 1 day and 1 week the values don’t include outlier beyond this range. We labeled the Y axis with the respective units used in our manuscript (G/l for COP, fl for MPV and the ratio for COP/MPV) and the title of each graph denote the respective marker. For further clarification, we added information on the units in the methods section.
The same is also true for the over simplified figure legend 2, which is difficult for readers to understand the information from these 9 figures.
Response: In our submitted version of the figure, we included lines to denote columns and rows, which are not present in this version. Our goal was to allow for easy readability of the figure, and we therefore added more information into the figure itself. (column title denotes the marker and row title denotes the timepoint). We found this figure representation to be the clearest arrangement. We added information on the Y axis in the figure legend 2. Maybe the editor can help with readability by adding lines to the figure for accentuation of columns and rows?
The authors may consider to decrease usage some of the abbreviations since they increase the difficulties for readers to understand their findings in the manuscript.
Response: We share this concern and reduced some of the abbreviations where possible.

Reviewer 2 Report
In this study the authors evaluate the prognostic significance of the perioperative (preoperative, at one day postoperative, and at 7 days postoperative) platelet count (COP), mean platelet volume (MPV) and the MPV/COP ratio in a series of surgically treated head and neck cancer patients. Additionally, they analyzed the individual postoperative trajectory of these indices and their association with prognosis. Overall, the work is well designed and written, and the conclusions are supported by the results. The main novelty is the prognostic value of the postoperative evolution of COP, since the relationship between COP and MPV with prognosis has already been the subject of other studies and meta-analysis.
The main limitation of the work, mentioned by the authors, is that it has not taken into account the comorbidities of patients, which can influence platelet levels, and in turn have been shown to have a significant influence on the prognosis of patients. Blood loss during surgery and the need for transfusion should also have been taken into account.
Specific comments
- The authors stated that all the patients were treated with curative surgery and postoperative RT. However, 32% of patients were classified as stage I-II disease. These patients could be treated with a single treatment modality; Why did they receive combination treatment?
- The method used to establish HPV status should be explained in the methods section. In results, the authors point out that detection by in situ hybridization was used, but it should be combined with the determination of p16 to distinguish active from non-active infection.
- It is indicated that the 8th TNM classification was used, but in the N classification only one patient was classified as N3, and N3a and N3b are not separated, which suggests that the tumors were actually classified following the 7th edition, as it seems unlikely that any patient had extranodal extension (N3b). The terms T classification and N classification should be used instead of T and N stage.
- Cut-off points were selected empirically. It seems that the selection of the cut-off points should have been better carried out by analyzing the ROC curves.
- The paragraph between lines 119 and 123 is redundant with the data in table 1, and could be deleted
Author Response
Reviewer 2:
In this study the authors evaluate the prognostic significance of the perioperative (preoperative, at one day postoperative, and at 7 days postoperative) platelet count (COP), mean platelet volume (MPV) and the MPV/COP ratio in a series of surgically treated head and neck cancer patients. Additionally, they analyzed the individual postoperative trajectory of these indices and their association with prognosis. Overall, the work is well designed and written, and the conclusions are supported by the results. The main novelty is the prognostic value of the postoperative evolution of COP, since the relationship between COP and MPV with prognosis has already been the subject of other studies and meta-analysis.
The main limitation of the work, mentioned by the authors, is that it has not taken into account the comorbidities of patients, which can influence platelet levels, and in turn have been shown to have a significant influence on the prognosis of patients. Blood loss during surgery and the need for transfusion should also have been taken into account.
Response: We also thank reviewer #2 for the time spent to read and review our manuscript. It is clear from the questions and comments, that the reviewer is clearly an expert in the field of head and neck cancer.
Specific comments
- The authors stated that all the patients were treated with curative surgery and postoperative RT. However, 32% of patients were classified as stage I-II disease. These patients could be treated with a single treatment modality; Why did they receive combination treatment?
Response: Again, thank you for this comment. All of our patients were treated based on the 7th or earlier edition of the AJCC guidelines. Thus, some of the patients included in this study (mostly HPV(+)) were then classified as a higher stage and therefore received adjuvant treatment according to our boards policy. For this study we have used the most recent edition of the AJCC guidelines and as a result several patients were down-staged.
- The method used to establish HPV status should be explained in the methods section. In results, the authors point out that detection by in situ hybridization was used, but it should be combined with the determination of p16 to distinguish active from non-active infection.
Response:
We added the information used for HPV detection in the methods section.
We additionally evaluated the p16 expression using IHC but did not use this data in this study. Results for HPV in situ hybridization (HPV) and p16 IHC staging (p16) were concordant in 98 of 116 patients (84%), while 2 patients were HPV +/p16- (2%) and 16 were HPV-/p16+ (14%). However, to determine transcriptionally active from non-active HPV-related oropharyngeal carcinoma, to our understanding, mRNA expression levels for E6/E7 would be necessary(1).
- It is indicated that the 8th TNM classification was used, but in the N classification only one patient was classified as N3, and N3a and N3b are not separated, which suggests that the tumors were actually classified following the 7th edition, as it seems unlikely that any patient had extranodal extension (N3b). The terms T classification and N classification should be used instead of T and N stage.
Response: We thank the reviewer for this thoughtful comment. We updated the dataset to the 8th Edition TNM- classification before analyzing the dataset, but oversaw the extranodal extension in 3 HPV negative patients. We updated this information in the manuscript. We furthermore changed T- and N- stage into T- and N- classification in the manuscript as suggested.
- Cut-off points were selected empirically. It seems that the selection of the cut-off points should have been better carried out by analyzing the ROC curves.
Response:
We choose empirical cut-offs at the respective median values because of the relatively small cohort. Using a cut-off defined by ROC analysis would require a validation cohort. Otherwise overfitting of the statistical model would most likely occur and therefore bias the results. All statistical analyses were performed or reviewed by a statistician (DD).
- The paragraph between lines 119 and 123 is redundant with the data in table 1, and could be deleted
Response:
We agree with the suggestion and deleted the paragraph for better readability of the manuscript.
A. C. Jung et al., Biological and clinical relevance of transcriptionally active human papillomavirus (HPV) infection in oropharynx squamous cell carcinoma. Int J Cancer 126, 1882-1894 (2010).
